# An Empirical Study of Graph Contrastive Learning

**Yanqiao Zhu**[1,2]    **Yichen Xu**[3]    **Qiang Liu**[1,2]    **Shu Wu**[1,2*]

[1]Center for Research on Intelligent Perception and Computing
National Laboratory of Pattern Recognition
Institute of Automation, Chinese Academy of Sciences
[2]School of Artificial Intelligence, University of Chinese Academy of Sciences
[3]School of Computer Science, Beijing University of Posts and Telecommunications
yanqiao.zhu@cripac.ia.ac.cn   linyxus@bupt.edu.cn   {qiang.liu,shu.wu}@nlpr.ia.ac.cn

## Abstract

Graph Contrastive Learning (GCL) establishes a new paradigm for learning graph representations without human annotations. Although remarkable progress has been witnessed recently, the success behind GCL is still left somewhat mysterious. In this work, we first identify several critical design considerations within a general GCL paradigm, including augmentation functions, contrasting modes, contrastive objectives, and negative mining strategies. Then, to understand the interplay of different GCL components, we conduct comprehensive, controlled experiments over benchmark tasks on datasets across various domains. Our empirical studies suggest a set of general receipts for effective GCL, e.g., simple topology augmentations that produce sparse graph views bring promising performance improvements; contrasting modes should be aligned with the granularities of end tasks. In addition, to foster future research and ease the implementation of GCL algorithms, we develop an easy-to-use library PyGCL, featuring modularized CL components, standardized evaluation, and experiment management. We envision this work to provide useful empirical evidence of effective GCL algorithms and offer several insights for future research.

## 1 Introduction

The past few years have witnessed rapid advances in Graph Neural Networks (GNNs) [1, 2], which have become the de facto framework for analyzing graph-structured data. As the most GNN models focus on (semi-)supervised learning, which requires access to abundant labels, recent trends in Self-Supervised Learning (SSL) have led to a proliferation of studies on learning graph representations without relying on human annotations. Among them, Contrastive Learning (CL) is a major area of interest and has already achieved comparable performance with supervised counterparts in many representation learning tasks [3–12].

Recently, remarkable progress has been made to adapt CL for the graph domain. A typical Graph Contrastive Learning (GCL) method constructs multiple graph views via stochastic augmentation of the input graph and then learns representations by contrasting positive samples against negative ones. For each node being an anchor instance, its positive samples are often chosen as the congruent representations in other views, while negatives are selected from other nodes within the given graph or other graphs within the batch. Although GCL has constituted a new paradigm of SSL in the graph domain and achieved promising results, recent studies [13–19] seem to resemble each other with very limited nuances from the methodological perspective. Moreover, most existing work only provides model-level evaluation and the contributing factors leading to the success of GCL still remain somewhat mysterious, which calls for a deeper understanding of different GCL components.

---

[*]To whom correspondence should be addressed.

35th Conference on Neural Information Processing Systems (NeurIPS 2021) Track on Datasets and Benchmarks.

Towards this end, we try to shed light on how these GCL algorithms succeeded through the lens of empirical evaluation of critical design considerations. We first propose a general contrastive paradigm that characterizes previous work in a limited design space of four design dimensions: (a) data augmentation functions, (b) contrasting modes, (c) contrastive objectives, and (d) negative mining strategies. Note that we include no model-specific design considerations such as the number of attention heads for graph attentive encoders. To the best that we are aware, these four dimensions cover a wide range of options that are representative in open literature.

Then, we systematically study the empirical performance of different design dimensions through controlled experiments over benchmark tasks on a set of datasets across a variety of domains. Through the empirical studies, we attempt to provide answers to the following questions:

- What is the most contributory component in an effective GCL algorithm?
- How do different design considerations affect the model performance?
- Do these design considerations favor certain types of data or end tasks?

It is noted that there have been several survey papers on self-supervised graph representation learning [20–22]. However, to the best of our knowledge, none of existing work provides rigorous empirical evidence on the impact of each component in GCL.

We summarize several key findings of the empirical study, which we hope could benefit the graph SSL community for developing future algorithms. Our experiments suggest a set of general recipes for effective GCL algorithms:

- GCL algorithms benefit the most from topology augmentation that produces sparse graph views. In addition, bi-level augmentation on both topology and feature levels further improves the performance.
- Overall, same-scale contrasting modes are desirable. The contrasting modes should also be chosen according to the granularity of downstream tasks.
- The InfoNCE objective obtains stable, consistent performance improvements under all settings yet requires a large number of negative samples.
- Several recently proposed negative-sample-free objectives have great potential for reducing computational burden without compromising on performance.
- Current negative mining strategies based on calculating embedding similarities bring limited performance improvements to GCL.

In addition, to foster future research, we develop PyGCL, an easy-to-use PyTorch framework, featuring commonly used, modularized GCL components, standardized evaluation, and experiment management. We hope the use of PyGCL will greatly relief the burden of comparing existing baselines and developing new algorithms. The PyGCL is open-sourced at `https://github.com/GraphCL/PyGCL`.

## 2 A General Paradigm of GCL and its Design Dimensions

### 2.1 Problem Formulation

Let $\mathcal{G} = (\mathcal{V}, \mathcal{E})$ denote a given graph, where $\mathcal{V} = \{v_i\}_{i=1}^N$, $\mathcal{E} \subseteq \mathcal{V} \times \mathcal{V}$ represent the node set and the edge set respectively. We further denote the feature matrix and the adjacency matrix as $\boldsymbol{X} \in \mathbb{R}^{N \times F}$ and $\boldsymbol{A} \in \{0, 1\}^{N \times N}$, where $\boldsymbol{x}_i \in \mathbb{R}^F$ is the feature of $v_i$ and $\boldsymbol{A}_{ij} = 1$ iff $(v_i, v_j) \in \mathcal{E}$. In the setting of unsupervised representation learning, there is no given class information of nodes or graphs during training. Our objective is to learn a GNN encoder $f(\cdot)$ receiving the graph features and structure as input and producing node embeddings in a low dimensionality. We denote $\boldsymbol{H} = f(\boldsymbol{X}, \boldsymbol{A}) \in \mathbb{R}^{N \times F'}$ as the learned representations of nodes, where $\boldsymbol{h}_i$ is the embedding of node $v_i$. For graph-oriented tasks, we can further obtain a graph-level representation $\boldsymbol{s} = r(\boldsymbol{H}) \in \mathbb{R}^{F'}$ of $\mathcal{G}$ that aggregates node-level embeddings. Note that the readout function $r(\cdot)$ might be a simple permutation-invariant function such as mean or sum pooling, or may be parameterized by a neural network. These representations can be used in downstream tasks, such as node/graph classification and community detection.

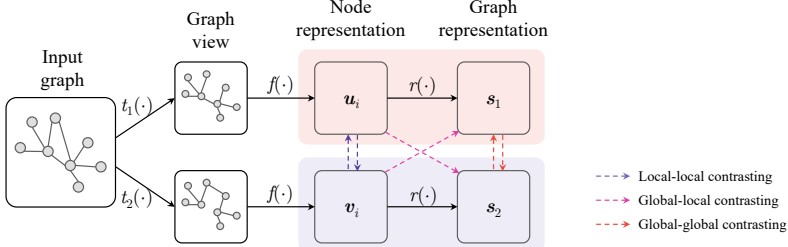

Figure 1: A general GCL model. At first, two graph views are generated via data augmentation functions. Then, the two graphs are fed into a shared graph neural network to learn representations, which are then optimized with a contrastive objective that pulls congruent representation pairs in the two views together while pushing others away. Additional negative mining techniques may be employed to improve the model performance.

## 2.2 A General Paradigm of GCL

We decompose GCL algorithms from four dimensions: (a) data augmentation functions, (b) contrastive mode, (c) contrastive objective, and (d) negative mining strategies. These four components constitute the design space of interest in this work.

At each iteration of training, we first perform stochastic **augmentation** to generate multiple graph views from the input graph. Specifically, we sample two augmentation functions $t_1, t_2 \sim \mathcal{T}$ to generate graph views $\widetilde{\mathcal{G}}_1 = t_1(\mathcal{G})$ and $\widetilde{\mathcal{G}}_2 = t_2(\mathcal{G})$, where $\mathcal{T}$ is the set of all possible transformation functions to be discussed in the next section. We then obtain node representations for the two views using a shared GNN encoder $f(\cdot)$, denoted by $\boldsymbol{U} = f(\widetilde{\boldsymbol{X}}_1, \widetilde{\boldsymbol{A}}_1)$ and $\boldsymbol{V} = f(\widetilde{\boldsymbol{X}}_2, \widetilde{\boldsymbol{A}}_2)$ respectively. Optionally, we may obtain graph representations for each graph view through a readout function $r(\cdot)$: $\boldsymbol{s}_1 = r(\boldsymbol{U})$ and $\boldsymbol{s}_2 = r(\boldsymbol{V})$.

For every node embedding $\boldsymbol{v}_i$ being the anchor instance, the **contrasting mode** specifies a positive set $\mathcal{P}(\boldsymbol{v}_i) = \{\boldsymbol{p}_i\}_{i=1}^P$ and a negative set $\mathcal{Q}(\boldsymbol{v}_i) = \{\boldsymbol{q}_i\}_{i=1}^Q$. In a pure unsupervised learning setting, we only consider congruent samples in each graph view; in other words, embeddings in the two augmented graph views corresponding to the same node or graph constitute the positive set. When label supervision is given, the positive set may be enlarged with samples belonging to the same class [23]. Moreover, we may employ **negative mining strategies** to improve the negative sample set by considering the relative similarity (i.e. the hardness) of negative samples. Finally, we use a **contrastive objective** $\mathcal{J}$ to score these specified positive and negative pairs.

### 2.2.1 Design Dimensions

In the following, we succinctly list implementations of these four design dimensions considered in this work. For details of each implementation, we refer readers of interest to Appendix F.

- **Data augmentation.** The purpose of data augmentation is to generate *congruent, identity-preserving positive samples* of the given graph. Most GCL work involves bi-level augmentation techniques: topology (structure) and feature augmentation.

  - **Topology augmentation:** (1) Edge Removing (ER), (2) Edge Adding (EA), (3) Edge Flipping (EF), (4) Node Dropping (ND), (5) Subgraph induced by Random Walks (RWS), (6) diffusion with Personalized PageRank (PPR), and (7) diffusion with Markov Diffusion Kernels (MDK).

  - **Feature augmentation:** (1) Feature Masking (FM) and (2) Feature Dropout (FD).

- **Contrasting modes.** For an anchor instance, contrasting modes determine the positive and negative sets at different granularities of the graph. In mainstream work, three contrasting modes are widely employed: (1) local-local CL and (2) global-global CL, which contrast embeddings at the same scale, and (3) global-local CL, which contrasts cross-scale embeddings. Note that the space of contrasting mode depends on downstream tasks. Only local-local and global-local CL are applicable for node datasets, where all three mode can be used for graph datasets.

Table 1: Summary of representative GCL models within the proposed paradigm.

| Method | Primary task | Topology augmentation | Feature augmentation | Contrasting mode | Dual branches? | Contrastive objective |
|---|---|---|---|---|---|---|
| DGI [13] | Node classification | — | — | Global-local | ✗ | JSD |
| GMI [15] | Node classification | — | — | Global-local | ✗ | SP-JSD |
| InfoGraph [19] | Graph classification | — | — | Global-local | ✗ | SP-JSD |
| MVGRL [14] | Node & graph classification | PPR | — | Global-local | ✓ | JSD |
| GCC [24] | Transfer learning | RWS | — | Local-local | ✗ | InfoNCE |
| GraphCL [16] | Graph classification | RWS/ND/EA/ED | FD | Global-global | ✓ | InfoNCE |
| GRACE [17] | Node classification | ER | MF | Local-local | ✓ | InfoNCE |
| GCA [18] | Node classification | ER | MF | Local-local | ✓ | InfoNCE |
| BGRL [25] | Node classification | ER | MF | Local-local | ✓ | BL |
| GBT [26] | Node classification | ER | MF | Local-local | ✓ | BT |

- **Contrastive objectives.** Contrastive objectives are used to measure the similarity of positive samples and the discrepancy between negatives. We consider the following objective functions that rely on negative samples: (1) Information Noice Contrastive Estimation (InfoNCE), (2) Jensen-Shannon Divergence (JSD), and (3) Triplet Margin loss (TM). Moreover, we also analyze the following objectives that eschew the need of explicit negative samples: (4) the Bootstrapping Latent loss (BL), (5) Barlow Twins (BT) loss, and (6) VICReg loss.

- **Negative mining strategies.** Recent work argues that CL algorithms benefits from hard negative samples (i.e. samples difficult to distinguish from an anchor instance). In this work, we consider the following four negative mining strategies: (1) Hard Negative Mixing (HNM), (2) Debiased Contrastive Learning (DCL), (3) Hardness-Biased Negative Mining (HBNM), and (4) Conditional Negative Mining (CNM).

### 2.2.2 Discussions on Representative GCL Methods

We give a brief summary of existing representative GCL methods as shown in Table 1 and discuss how they fit into our proposed paradigm. We note that negative mining strategies have received scant attention in current GCL literature and thus are omitted in the table.

**Dual branches vs. single branch.** We notice that most work leverages a dual-branch architecture following SimCLR [8] that augments the original graph twice to form two views and designates positive samples across two views. For some global-local CL methods like DGI [13] and GMI [15], they employ an architecture with only one branch. In this case, negative samples are obtained by corrupting the original graph. Different from the aforementioned *augmentation* schemes that generate congruent pairs to model *the joint distribution* of positive pairs, we resort to the term *corruption functions*, which approximate *the product of marginals*.

**Stronger augmentation.** Unlike GRACE [17] and GraphCL [16] that employ uniform edge/feature perturbation, GCA [18] proposes to perform adaptive augmentation based on importance scores of edges and features. In this work, to involve less hyperparameters as possible, we focus on "uniform" augmentation schemes only.

**Variants of contrasting modes.** GMI [15] extends DGI [13] by further considering the agreement between raw node/edge features and node/edge representations. Because it requires much more computational resources, our experiments exclude this implementation of contrasting mode. In addition, there are several recent methods [27, 28] involve contrasting between local/global and *context* representations, which are usually derived from graph clustering algorithms. Considering the generality of the experiments, we shall leave studying these variants as a future direction.

## 3 Empirical Studies

In the following section, we first introduce experimental configurations and then summarize the results and observations regarding each particular component in the proposed paradigm. We refer interested readers to Appendix B for more information on the evaluation protocols and implementations.

**Evaluation configurations.** We conduct experiments on a variety of medium- to large-scale datasets widely used in literature, ranging from academic networks to chemistry molecules. For fair comparison, we closely follow previous studies on datasets preprocessing [13, 16–19, 25, 29, 30]. We mainly

Table 2: Statistics of datasets used for unsupervised learning experiments.

| Dataset | Domain | Task | #Graphs | Avg. #nodes | Avg. #edges | #Features | #Classes |
|---------|--------|------|---------|-------------|-------------|-----------|----------|
| Wiki | Knowledge base | Unsupervised node classification | 1 | 11,701 | 216,123 | 300 | 10 |
| Computer | Social networks | | 1 | 13,752 | 245,861 | 767 | 10 |
| CS | Citation networks | | 1 | 18,333 | 81,894 | 6,805 | 15 |
| Physics | Citation networks | | 1 | 34,493 | 247,962 | 8,415 | 5 |
| NCI1 | Biochemical molecules | Unsupervised graph classification | 4110 | 29.87 | 32.30 | — | 2 |
| PROTEINS | Bioinformatics | | 1,133 | 39.06 | 72.82 | 29 | 2 |
| IMDB-M | Social networks | | 1,500 | 13.00 | 65.94 | — | 3 |
| COLLAB | Social networks | | 5,000 | 74.49 | 2457.78 | — | 3 |

evaluate models with different design considerations on two essential tasks: (1) unsupervised node classification and (2) unsupervised graph classification. For all experiments, we follow the linear evaluation scheme used by DGI [2]. Particularly, the models are first trained in an unsupervised manner and the resulting embeddings are fed into a linear classifier to fit the labeled data. We evaluate each model using ten random splits and report the averaged accuracies (%) as well as standard deviation scores.

**Datasets.** We conduct experiments on a variety of medium-scale datasets widely used in open literature, ranging from academic networks to chemistry molecular datasets, including Wiki-CS (Wiki) [31], Amazon-Computer (Computer), Coauthor-CS (CS), and Coauthor-Physics (Physics) [29] for node classification and NCI1 [32], PROTEINS-full (PROTEINS) [33], IMDB-MULTI (IMDB-M), and COLLAB [34] for graph classification. The statistics of all datasets is summarized in Table 2.

**Implementation details.** We choose Graph Convolutional Networks (GCNs) [1] as the encoder for all node tasks and use Graph Isomorphism Networks (GINs) [35] for all graph tasks. For projector functions, we utilize an additional MultiLayer Perceptron (MLP) model. To ensure convincing experiments and observations, we first perform an exhaustive search over the entire design space. Then, we select and report representative results to reveal common, useful practices. With the aim of conducting controlled experiments, we fix as many variables, e.g., the GNN encoder architecture, embedding dimensions, the number of epochs, and activation functions, as possible for every dataset.

## 3.1 Data Augmentation

⋄ Augmentation: *eval*    ⋄ Contrasting mode: L–L    ⋄ Objective: InfoNCE    ⋄ Negative mining strategy: None

We first investigate how data augmentation affectss the performance of GCL. Specifically, we apply different data augmentation functions to generate two views, leverage the InfoNCE objective, and contrast local-local (node) representations. Except for specific augmentation functions used, all other settings are kept the same. Below, we first report performance on different topology and feature augmentation functions. Then, we examine the use of compositional data augmentation schemes: (1) structure- and feature-level augmentations and (2) deterministic plus stochastic augmentations.

**Observation 1. Topology augmentation greatly affects model performance. Augmentation functions that produce sparser graphs generally lead to better performance.**

Table 3 summarizes the results of employing different topology and feature augmentations. It is evident that the performance of GCL is highly dependent on the choice of topology augmentation functions. We observe that models that remove edges (ER, MDK, ND, PPR, and RWS), compared to models that add edges (EA), in general achieve better performance, which suggests that augmentation functions produce *sparser graph views* are preferable. We also find that Random Walk Sampling (RWS) achieves better performance on node datasets, while Node Dropping (ND) favors graph tasks. Although random walk sampling is able to better extract local structural patterns, the graph datasets used in our study are of relatively small scales (< 500 nodes per graph). Therefore, we suspect that these random-walk-based sampling strategies may be confined and a simple node dropping scheme generally outperforms other augmentations on graph-level tasks.

To see how sparsity of the resulting views affect the performance, we further conduct sensitivity analysis on three models with ND, ER, and EA augmentation respectively by varying the dropping/adding probabilities on the CS dataset. The prediction accuracy along with the total number of edges in the

Table 3: Classification performance with different topology and feature augmentation. The best performance is highlighted in boldface and the second-to-best underlined. OOM indicates Out-Of-Memory on a 24GB GPU.

| | Aug. | Node | | | | Graph | | | |
|---|---|---|---|---|---|---|---|---|---|
| | | Wiki | CS | Physics | Computer | NCI1 | PROTEINS | IMDB-M | COLLAB |
| | None | 68.52±0.39 | 90.76±0.05 | 93.69±0.73 | 80.62±0.62 | 58.49±2.21 | 70.94±1.13 | 45.07±1.70 | 66.21±0.92 |
| Topo. | EA | 72.65±0.43 | 92.73±0.10 | 94.77±0.05 | 83.40±0.64 | 70.80±0.55 | 71.17±0.63 | 44.80±1.43 | 68.12±0.63 |
| | ER | 76.38±0.21 | 92.83±0.17 | 95.21±0.05 | 87.84±0.76 | 73.03±0.48 | 72.55±0.11 | 45.17±1.64 | 68.13±0.82 |
| | EF | 74.10±0.67 | 92.99±0.15 | 94.88±0.06 | 86.68±0.73 | 73.95±0.49 | 70.64±1.67 | 44.15±1.21 | 67.92±0.93 |
| | ND | 77.47±0.32 | 92.81±0.08 | 95.99±0.12 | 87.01±0.54 | 72.12±1.38 | 72.54±0.43 | 47.03±1.14 | 70.73±0.78 |
| | PPR | 69.28±0.22 | 92.25±0.07 | OOM | 85.06±0.53 | 58.70±0.51 | 71.69±1.12 | 45.27±0.85 | 68.51±0.67 |
| | MKD | 69.87±0.12 | 92.62±0.14 | OOM | 82.46±0.58 | 57.21±0.31 | 71.31±0.11 | 45.07±1.16 | 68.09±0.88 |
| | RWS | 76.74±0.20 | 93.48±0.08 | 95.04±0.11 | 87.60±0.63 | 75.11±1.14 | 71.79±0.82 | 44.95±0.82 | 70.85±0.89 |
| Feat. | FM | 76.74±0.34 | 91.55±0.11 | 94.12±0.21 | 85.05±0.51 | 64.87±0.36 | 71.35±0.79 | 45.36±1.68 | 70.52±0.35 |
| | FD | 76.68±0.16 | 91.83±0.08 | 94.20±0.16 | 84.93±0.46 | 63.21±0.51 | 71.60±1.61 | 46.44±0.96 | 70.69±1.33 |

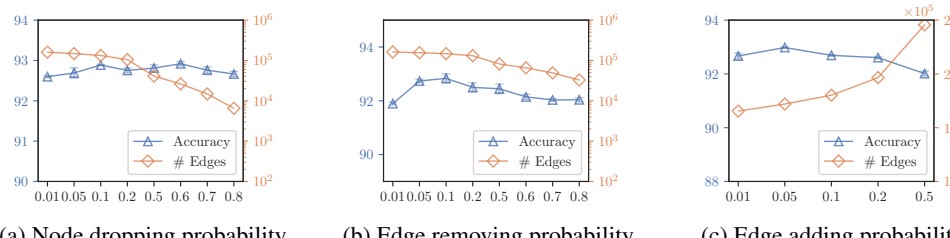

(a) Node dropping probability     (b) Edge removing probability     (c) Edge adding probability

Figure 2: Sensitivity analysis with various topology augmentation probabilities on the CS dataset.

produced graphs is plotted in Figure 2. From Figures 2a and 2b, we observe that model performance improves as more nodes or edges are dropped and degenerates when the removal probability is overly high. As seen in Figure 2c, the performance of EA augmentation downgrades greatly when more edges are added. In general, these results accord with our observations that many real-world graphs are inherently sparse [36, 37]. When too many edges are added, they connect nodes that are semantically unrelated, bringing noise to the generated graph view and thus deteriorating the quality of learned embeddings.

**Observation 2. Feature augmentation brings additional benefits to GCL. Compositional augmentation at both structure and attribute level benefits GCL the most.**

From Table 3, we observe that the performance of models employing feature augmentation solely is inferior to that use topology augmentation but still higher than the baseline performance. The

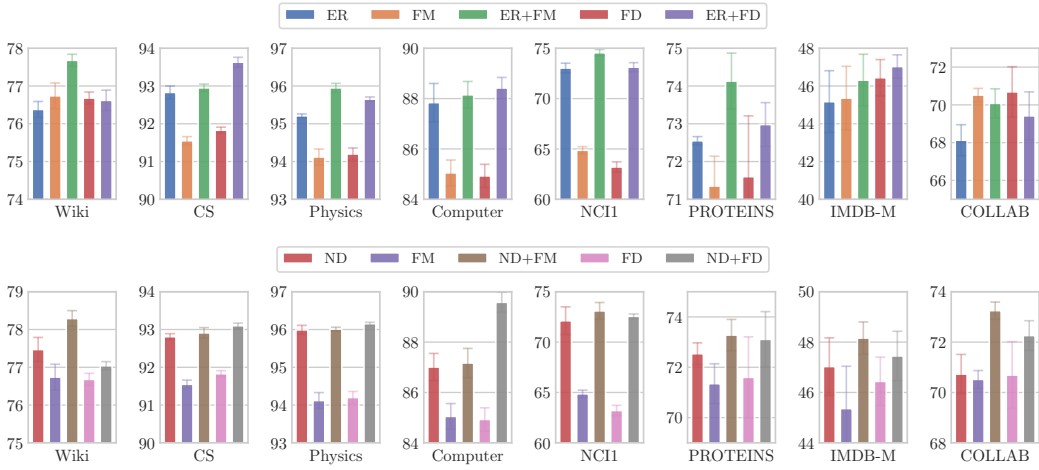

Figure 3: Performance on models using compositional augmentation at topology and feature levels.

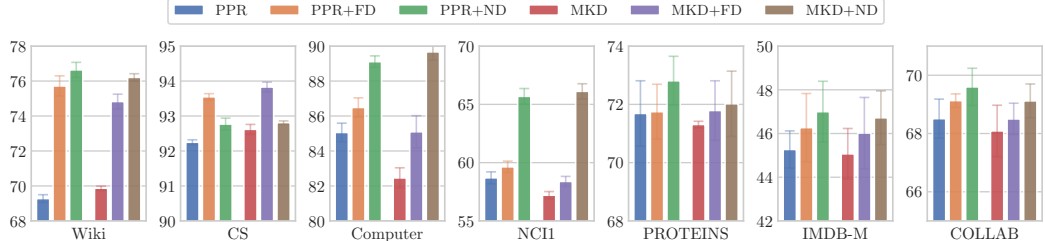

Figure 4: Performance on models using composition of stochastic and deterministic augmentation.

improvements brought by the two feature augmentation schemes could be explained from the fact that FM and FD resemble applying the dropout [38] technique on the input layer. Also, we see that in general FM slightly outperforms FD, which suggests the use of a shared feature mask for all node features, though their performance difference is not that significant.

Furthermore, Figure 3 presents the results in the case of hybrid augmentation at both topology and feature levels. It is clear that the use of feature augmentation in addition to structure augmentation benefits GCL, demonstrating that both topology and structures are important for learning graph representations.

**Observation 3. Deterministic augmentation schemes should be accompanied by stochastic augmentation.**

In Table 3, we find that using two deterministic augmentation functions PPR and MDK independently does not always result in promising performance. We also find that prior studies [14] usually leverage stochastic augmentation (e.g., node dropping) after deterministic augmentation like PPR. Therefore, we utilize another set of joint augmentation of stochastic and deterministic augmentation functions, where the performance is summarized in Figure 4. It is seen that the joint scheme improves the vanilla deterministic augmentation by a large margin. Recall that our contrastive objective is essentially aimed to discriminate between samples from the data distribution and samples from some noise distributions [39, 40]. Therefore, a stochastic augmentation scheme is needed to better approximate that noise distribution.

## 3.2  Contrasting Modes and Contrastive Objectives

◇ Augmentation: ND + FM    ◇ Contrasting mode: *eval*    ◇ Objective: *eval*    ◇ Negative mining strategy: None

The next experiments are concerned with how contrasting modes and contrastive objectives affect the model performance. We train the model with different contrasting modes and contrastive objectives, with the topology augmentation set to ND and the feature augmentation to FM. Except that embedding sizes are fixed, other hyperparameters are tuned to obtain the best performance under each experiment for fair comparison.

We first experiment with negative-sample-based contrastive objectives: InfoNCE, JSD, and TM. Besides contrastive objectives that rely on negative samples, we also investigate three negative-sample-free objectives: BL, BT, and VICReg. Table 4 presents the experimental results on unsupervised classification tasks. It should be noted that BT and VICReg losses need to compute the covariance of latent embedding vectors and thus are not compatible with the global-local mode.

**Observation 4. Same-scale contrasting generally performs better. Downstream tasks of different granularities favor different contrasting modes.**

What stands out from the table is that contrasting local-local pairs achieves the best performance on node-level classification, while the global-global mode performs better on graph-level tasks. This suggests us to use same-scale contrasting modes, which is consistent with the practices used in recent literature [5, 8, 16–18, 41–44]. A possible explanation for this result is that, in global-local contrasting, all node embeddings within the graph constitute the positive samples for each graph embedding. In other words, the global-local mode pulls every node-graph pair together in the embedding space, which may lead to suboptimal performance in downstream tasks.

Table 4: Performance with different objectives and contrasting modes. L–L, G–L, and G–G denote local-local, global-local, and global-global contrasting modes. The best results for objectives (row-wise) and contrasting modes (column-wise) are highlighted in boldface and underline respectively.

(a) Unsupervised node classification

| Obj. | Wiki | | CS | | Physics | | Computer | |
|---|---|---|---|---|---|---|---|---|
| | L–L | G–L | L–L | G–L | L–L | G–L | L–L | G–L |
| InfoNCE | **79.09±0.15** | 77.73±0.94 | **92.45±0.83** | 90.60±0.06 | **95.95±0.92** | 93.23±0.96 | **88.15±0.59** | 76.24±0.93 |
| JSD | **78.83±0.95** | 78.71±0.19 | 92.18±1.00 | 91.31±0.62 | **94.32±0.28** | 94.12±0.04 | 82.02±0.76 | 78.27±0.05 |
| TM | **78.42±0.88** | 76.53±0.85 | **91.91±0.31** | 90.11±0.61 | **94.11±0.60** | 92.78±0.12 | 69.67±0.88 | 76.38±0.75 |
| BL | 76.83±0.80 | 75.34±0.43 | 93.10±0.94 | 88.55±0.43 | 94.81±0.98 | 94.09±0.83 | **87.79±0.94** | 85.43±0.23 |
| BT | 80.41±0.15 | — | **94.16±0.02** | — | **96.55±0.12** | — | 86.86±0.97 | — |
| VICReg | **80.79±0.12** | — | 93.46±0.08 | — | 95.59±0.23 | — | 86.39±0.32 | — |

(b) Unsupervised graph classification

| Obj. | NCI1 | | | PROTEINS | | | IMDB-M | | | COLLAB | | |
|---|---|---|---|---|---|---|---|---|---|---|---|---|
| | L–L | G–L | G–G | L–L | G–L | G–G | L–L | G–L | G–G | L–L | G–L | G–G |
| InfoNCE | 73.10±0.83 | 72.35±0.21 | **73.95±0.89** | 73.28±0.62 | 71.57±0.92 | **75.73±0.09** | 48.16±0.64 | 47.36±0.48 | **49.69±0.44** | 73.25±0.34 | 70.92±0.22 | **73.72±0.12** |
| JSD | 73.56±0.32 | 73.29±0.31 | 70.93±0.17 | **73.88±0.31** | 73.15±0.42 | 73.67±0.45 | 48.31±1.17 | **48.61±1.21** | 48.31±1.35 | 70.40±0.31 | **72.62±0.35** | 71.60±0.32 |
| TM | 72.43±0.21 | 71.21±0.19 | 72.31±0.22 | 72.17±0.51 | 72.13±1.48 | **73.78±0.47** | 48.38±0.20 | 47.75±1.24 | **48.58±0.62** | 68.85±0.45 | 69.47±0.20 | **72.97±0.47** |
| BL | **77.22±0.13** | 75.97±0.23 | 76.70±0.31 | **77.75±0.43** | 77.32±0.21 | **78.17±0.59** | 54.64±0.43 | 54.21±1.01 | **55.32±0.21** | **73.95±0.25** | 73.35±0.24 | **74.92±0.33** |
| BT | 72.49±0.22 | — | 70.53±1.11 | 74.87±0.68 | — | 74.38±0.56 | 48.50±0.65 | — | 49.53±0.42 | 71.70±0.53 | — | 73.00±0.42 |
| VICReg | 72.31±0.34 | — | 71.60±0.36 | 74.61±1.15 | — | 74.38±0.57 | 46.75±1.47 | — | 50.28±0.55 | 68.88±0.34 | — | 72.50±0.31 |

Furthermore, the contrasting mode should be chosen according to the granularity of end task, i.e. local-local for node-level tasks and global-global for graph-level tasks. In accordance with the presented results, recent studies [45–47] have made a similar finding for learning visual representations. They demonstrate that being pretrained on instance-level pretext tasks (i.e. contrasting image-level embeddings in the same batch), current CL models achieve suboptimal performance in fine-grained downstream tasks, e.g., semantic segment that requires pixel-level details.

**Observation 5. Among negative-sample-based objectives, the use of InfoNCE objective leads to consistent improvements across all settings.**

Table 4a indicates that InfoNCE achieves the best performance among contrastive objectives that need negative samples, which is demonstrated to be effective by many recent methods [4, 7, 8, 48, 49].

Among these negative-sampling-based objectives, recent studies have already revealed that the InfoNCE loss has an intrinsic ability to perform hard negative sampling [23], which may explain its superior performance compared to other objectives. Particularly, one very recent study in computer vision [50] shows that the use of a temperature parameter $\tau$ in the InfoNCE objective acts as an adjustment factor to exert penalties on hard negative samples. To verify this on GCL, we further conduct sensitivity analysis on this temperature parameter on both node (CS) and graph (IMDB-M) datasets as shown in Figure 5. We observe that with the increase of $\tau$, the performance improves at first and downgrades later, with not much performance fluctuation.

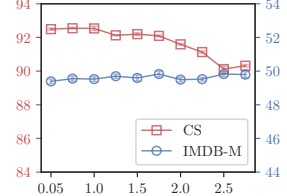

Figure 5: Sensitivity analysis of the parameter $\tau$ in the InfoNCE objective.

According to Wang and Liu [50], the InfoNCE objective pays less attention to hard negatives as $\tau$ increases. This hardness-aware behavior demonstrates the importance of *striking a balance* between separation of hardest negative samples ($\tau \to 0^+$) and global uniformity ($\tau \to \infty$) on GCL.

**Observation 6. Bootstrapping Latent and Barlow Twins losses obtain promising performance on par with their negative-sample-based objectives yet reduce the computational burden.**

Surprisingly, we find the performance obtained by employing these negative-sample-free objectives sometimes even surpasses their negative-sample-based counterparts, which suggests a promising future direction of finding efficient solutions free of negative samples for GCL. As opposed to negative-sample-based objectives, the BL, BT, and VICReg losses eschew the need of explicit negative samples and thus greatly reduce the computational burden. To see this, we summarize the memory consumption in Table 5, from which we clearly observe that these three losses use much less memory than other objectives without negative samples.

Table 5: Memory usage (MB) on the PROTEINS dataset of different contrastive objectives.

| Obj. | L–L | G–L | G–G |
|---|---|---|---|
| InfoNCE | 6,311 | 2,977 | 2,271 |
| JSD | 6,309 | 2,845 | 2,269 |
| TM | 6,271 | 2,977 | 2,269 |
| BL | 2,235 | 2,247 | 2,187 |
| BT | 2,419 | — | 2,201 |
| VICReg | 2,465 | — | 2,232 |

## 3.3 Negative Mining Strategies

◇ Augmentation: ND + FM ◇ Contrasting mode: L–L ◇ Objective: InfoNCE ◇ Negative mining strategy: *eval*

We firstly probe the explicit use of negative mining strategies on top of contrastive objectives, which essentially measure similarity (i.e. hardness) of each negative pair and upweights hard negative samples. Among the examined negative mining strategies, DCL develops debasing terms to select truly negative samples so as to avoid contrasting same-label instances; other strategies propose to upweight hard negative samples (points that are difficult to distinguish from an anchor) and remove easy ones that are less informative to improve the discriminative power of the GCL model.

**Observation 7. Existing negative mining techniques based on calculating embedding similarities bring limited benefit to GCL.**

We train four models on three node classification datasets (due to out-of-memory error on Physics) using the local-local mode with the InfoNCE objective. The performance comparison of different

negative mining strategies is shown in Figure 6. It is apparent that the four negative mining strategies studied in this experiment bring limited improvements to GCL. Although slight improvements can be observed under certain hyperparameter configurations, which proves the explicit modeling of hard negatives is useful for GCL, their performance in general rarely matches the best performance reported with other contrastive objectives (cf. Table 4a).

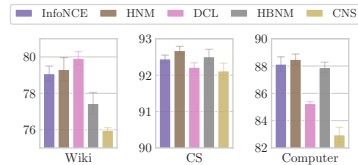

Figure 6: Performance of negative mining strategies.

We note that in existing formulation of negative sampling techniques, the sample hardness is measured by inner product of sample embeddings. Since we are in a fully unsupervised setting, no label (i.e. class) information can be accessed during training. Under existing contrasting modes, for one anchor sample, the contrastive objective pushes all different representations away, *irrespective of their real semantic relations*. What is even worse, most GNN models tend to produce similar embeddings for neighboring nodes regardless of semantic classes [51–54], which may further bias the selection of hard negatives. Therefore, we argue that there is disagreement between semantic similarity and example hardness. By selecting hard negative samples merely according to similarity measure of embeddings, these hard negatives are potentially *positive samples*, which produces adverse learning signals to the contrastive objective. For an illustrative example and more detailed discussions, please refer to Appendix D. We find that our discovery reminiscent to one very recent study in visual CL [50], which recommends an adaptive scheduling scheme for the temperature parameter when using InfoNCE as the contrastive objective, such that *hard but false negatives* could be tolerated as the training progresses.

## 3.4 Additional Experiments

We perform additional experiments on large-scale datasets and ablation studies on the batch normalization module used in the bootstrapping latent loss. Due to space constraints, we briefly discuss our findings as follows. Readers of interest may refer to Appendix C for detailed discussions.

**More evaluation tasks and metrics on large-scale datasets with edge features.** Currently, since the experiments are limited to medium-scale datasets, we conduct a further study assessing how existing GCL work scales to large-scale graphs using two open graph benchmarks [55] datasets. In the experiments, we examine the performance of different data augmentation due to the presence of edge features, contrasting modes, and contrastive objectives on binary graph classification and graph regression, which are measured in terms of AUC-ROC and MAE, respectively. It is shown that the experimental results are still in keeping with observations made in the main texts.

**Ablation studies on the Batch Normalization module for the Bootstrapping Latent loss.** Previous work [56] hypothesizes that the use of Batch Normalization (BN) in existing work compensates for improper initialization for contrastive learning models, instead of introducing implicit negative samples. To see its impact on GCL performance, we perform ablation studies by using BN or not in three critical components in the BL loss, i.e. the GNN encoder, the projector, and the predictor, on the node classification task. The results show that using BN in the GNN encoder *solely* is almost sufficient to obtain promising performance.

# 4 Conclusion, Limitations, and Outlook

In this paper, we first present a taxonomy for GCL, where we categorize existing work from four aspects: data augmentations, contrasting modes, contrastive objective, and negative mining strategies. Then, we analyze the design choices for each GCL component by conducting extensive empirical studies over a comprehensive set of benchmarking tasks and datasets. Our rigorous empirical study reveal several interesting findings of GCL components that may be helpful for developing future algorithms. We also provide an open-sourced PyTorch-based library PyGCL to facilitate the implementation of GCL models. While GCL has already demonstrated strong empirical performance across a variety of downstream tasks, it is still in its infancy with many challenges left widely open. We hope our work provides several practical guidelines for future research in this vigorous field.

Due to limited space, some limitations of our work need to be acknowledged.

- **Limited design considerations.** In this work, we consider limited design considerations, namely four design dimensions. An issue that is not addressed in this study is what role do many other model-specific factors, e.g., whether to employ a projection head in the InfoNCE objective and what graph encoders should be employed, play in GCL.
- **Limited downstream tasks.** Our empirical study only includes experiments on node- and graph-level classification and graph-level regression; a boarder range of downstream tasks of different granularities, e.g., link prediction and community detection, may be beneficial to draw more convincing conclusions.
- **Lack of theoretical justification.** Our work only presents empirical studies which has thrown up many questions in need of further theoretical justification for better understanding the underlying mechanisms of GCL, for instance, the performance guarantees of certain contrasting modes and how to appropriately measure and select hard negative samples for contrastive objectives in the graph domain.

Our empirical findings also suggest several future directions that may be helpful for fully unleashing the power of GCL.

- **Towards automated augmentation.** We understand that topology augmentation is of paramount importance to GCL. However, existing work leverages manually designed ad-hoc augmentation strategies, which may result in suboptimal performance. Recent studies in graph structure learning pave a principled way to learn optimal structures of graph-structured data [37], which we argue could be used for automatically learn augmentation functions suitable for GCL pretext tasks.
- **Understanding the performance gap between pretext and downstream tasks.** We empirically demonstrate the correlation between the choice of end tasks and contrastive objectives, yet calls for a thorough understanding for the performance gap between pretext and downstream tasks. We have found that there is some progress in this regard [57], but it is far from fully explored.
- **Structure-aware negative sampling.** Unlike in computer vision fields, similar visual features may naturally correlate to closer semantic categories, measuring the hardness through embedding similarities in graph-structured data is more difficult. A series of earlier work in network embedding proposes solutions from structural aspects [58–60]. However, how to integrate rich structure information for modeling better negative distributions for GCL is still left unexplored.

## Acknowledgments and Disclosure of Funding

The authors would like to thank Yuning You and Liang Zeng for their fruitful discussions on the manuscript. The authors would also like to thank anonymous reviewers for their helpful feedback. This work is supported by National Natural Science Foundation of China (61772528 and U19B2038).

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
