# OpenReview forum: "An Empirical Study of Graph Contrastive Learning"
_NeurIPS.cc/2021/Track/Datasets_and_Benchmarks/Round2 — NeurIPS 2021 Datasets and Benchmarks Track (Round 2)_

### Official Review · Reviewer_PaxT · 2021-09-10
**An extensive study on GCL**

**Rating:** 6
**Confidence:** 2
**Clarity:** The paper is well written and in gene…

**Strengths:**

1. A good breakdown of GCL methods into design components.
2. Observations are insightful and guide designs of GCL methods.
3. Clear discussion of drawbacks and future works.
4. Documentation is clear.

**Weaknesses:**

1. Some details are missing, which makes understanding this paper hard. In the main paper, the authors briefly describe the design dimensions, but no details are stated. This makes understanding hard. For example, in the **data augmentations**, I find it hard to understand the difference between FM and FD. Also, in the negative mining strategies, few details are given (even in the appendix).

**Additional Feedback:**

Section 2.1: Jenson-Shannon Divergence instead of Jason-Shannon Divergence.

**Correctness:**

I think the observations and insights derived are correct and justified by experiments.

**Documentation:**

The authors provide a link to the github repo and also provide a clear README. I think the documentation is OK.

**Ethics:**

No.

**Relation To Prior Work:**

I am not highly familiar with GCL. From my perspective, I think the discussion is OK.

**Summary And Contributions:**

This paper performs an extensive study on graph contrastive learning (GCL). The authors decompose the design components of GCL into four aspects, augmentation, contrasting modes, objectives, and negative sample mining. The authors further perform extensive experiments based on these four aspects.

The contributions of this paper include:
1. A good breakdown of the design choices in GCL into four major aspects.
2. Extensive experiments on different datasets and tasks (node-level and graph-level).

---

### Official Review · Reviewer_rwpd · 2021-09-18
**Good paper, but some clarification needed**

**Rating:** 7
**Confidence:** 4

**Strengths:**

+ usage of popular benchmark datasets
+ extensive analysis according to four taxonomy dimensions
+ multiple downstream tasks
+ clear observations/conclusions that can be used in future GCL research
+ the experimental code is available and the open-sourced PyGCL library allows to easily build custom GCL models

**Weaknesses:**

**W1. Motivation for experimental setup**

For each experimental setup it would be beneficial to explain why the authors choose such a setting, e.g. in "3.1 Data Augmentations" why did the authors specifically use the InfoNCE objective and not another one? Maybe another objective would lead to different observations?

Although subsection "3.2 Contrasting Modes and Contrastive Objectives" shows that InfoNCE is the best, we do not know whether this is due to the usage of specific data augmentations, which were the best performing in the previous experiment (again using the InfoNCE objective).

I encourage the authors to address the experimental scenarios design choices, at least in the Appendix as there are space constraints for the main paper.

**W2. Missing details / clarity**
- Experiment "3.3 Negative Mining Strategies" lacks information about which augmentations were used
- In the main paper, there is no information about the used backbone GNN, optimization procedure or other training details; please redirect the reader directly to appendix B.
- I would suggest that the authors include a standardized way to denote the setup in each experiment, e.g. in the first line of each experiment/subsection I would include the following summary line (or something similar), where you explicitly denote which part is *evaluated* and which ones are fixed. This should allow saving some space, compared to textual descriptions.
`
**Augmentions:** ND + FM | **Contrasting mode:** L-L  |  **Contrasting objective:** InfoNCE |  **Negative mining strategy:** *evaluated*
`

**W3. Comparison protocol**

In appendix B, the authors mention that in every experiment they use a grid search to find the best GNN model architecture. I am not fully convinced that such a strategy leads to meaningful results. In my opinion, it would be better to use the same architecture across scenarios, e.g. when checking the contrastive objective for a particular dataset, you should use the exact same GNN encoder architecture, embedding dimension, learning rates, etc. In the current case, there are too many degrees of freedom to get consistent results.

EDIT: Clarifications for the concerns were provided by the authors


**Additional Feedback:**

Instead of leaving it in the appendix, more emphasis should be put on negative-sample-free methods, as negative sampling can be too expensive for large-scale real-world datasets.

I am willing to increase my score after clarifications from the authors.

**Clarity:**

The paper is well written and easy to follow. There are a few minor grammar mistakes, but they should be easy to fix. I would suggest another proofreading round before the final revision.

**Correctness:**

I have some doubts about sticking mostly to the InfoNCE objective. Otherwise, the experiments are well performed, including the popular linear evaluation scheme, as proposed in DGI and used in many follow-up papers.

**Documentation:**

The attached codebase and appendix include all necessary steps and scripts to reproduce the reported results. Although, the authors could write a single script to launch all experiments, or use some dedicated library/tool to maintain reproducible experimental pipelines, e.g. DVC. The PyGCL code is also available.

**Ethics:**

No ethical concerns I can think of.

**Relation To Prior Work:**

The paper considers multiple GCL methods, including older ones and the most recent approaches. The authors also refer to other self-supervised graph learning studies and denote the differences to their work.

**Summary And Contributions:**

The paper studies graph contrastive learning approaches and performs an empirical comparison and analysis. The authors introduce a taxonomy to categorize existing methods according to four dimensions: augmentation functions, contrasting modes, contrastive objectives, and negative mining strategies. Thanks to such problem decomposition, the authors can experiment with different components of GCL methods and draw several conclusions for future GCL research. The authors also publish an open-source library PyGCL, which allows building graph contrastive learning methods using various ready-to-use components (aligned with the four taxonomy dimensions).

---

### Official Review · Reviewer_PZ7S · 2021-09-19
**Review of "an empirical study of graph contrastive learning"**

**Rating:** 7
**Confidence:** 2

**Strengths:**

The experiments are extensive; they cover the design dimensions that the authors have set out to explore fully. Methods that generally work the best has been identified for each design dimension, which I think would be very useful for anyone looking to apply GCL algorithms for their own problem setting. The authors have also identified directions that require further research, which are summarized at the end of Section 4.

**Weaknesses:**

Some augmentation methods and contrasting modes have been left out due to computational limitations (as mentioned in Section 2.2). Please see my comments for correctness as well. Other weaknesses have already been acknowledged by the authors in Section 4.

**Additional Feedback:**

I think the performance of feature augmentations alone (FM/FD) should be plotted in Figure 3 as well (so that it is easier to see hybrid augmentations perform the best compared with all non-hybrid options). Similarly, the performance of stochastic augmentations alone (FD/ND) can be plotted in Figure 4.

**Clarity:**

The paper is very well written; each design dimensions is defined clearly, experiments are summarized well through six observations (with the exception of Observation 1's inaccuracy), and limitations and future directions are outlined clearly in Section 4.

**Correctness:**

I believe Observation 1 is not correct. MDK and PPR are edge-removing augmentation methods yet they generally perform worse than EA (PROTEINS and IMDB-M are the only two environments where both MDK and PPR outperform EA). I think the results presented in Table 2 are still interesting but Observation 1 needs to be either removed or rephrased (and the text accompanying it needs to be revised as well).



**Documentation:**

The source code is made available online.

**Ethics:**

There are no ethical concerns.

**Relation To Prior Work:**

The prior work is summarized in Table 1 using the proposed way of decomposing GCL methods into four design dimensions. I think this unified view of the prior work is a strength.

**Summary And Contributions:**

The authors first characterize GCL models in terms of four design dimensions, namely data augmentations, contrasting modes, contrastive objectives, and negative mining strategies. Then, they empirically evaluate how each design decision impacts the performance in downstream tasks in a systematic manner.

---

### Official Review · Reviewer_wnP1 · 2021-09-19
**Round 2 Review for An Empirical Study of Graph Contrastive Learning**

**Rating:** 4
**Confidence:** 4

**Strengths:**

The demonstration that negative mining strategies generally don't improve performance and that InfoNCE is a consistent loss function for developers to start with is useful to the GNN community. In general, the authors conduct a large number of experiments across a variety of conditions in a standardized way. The field is quite diffuse and even though some newer stop grad methods are not included, these results are useful for researchers.

**Weaknesses:**

Generally, the design space is too large for the approach the authors take to study it and they don't have an argument that justifies that they can make general claims. A particularly concerning discrepancy is that they only work with graphs that have fewer than 500 nodes. There are many publicly available medium to large scale graphs available as benchmarks so the data examined should at least be representative of the current state of research. There are some interesting results, e.g., that augmentations greatly improve node but not graph classification, that the authors don't comment on and in so omitting such results, fail to point out interesting trends in the field, e.g., that the design space also depends on the task.

**Additional Feedback:**

Given the authors say the InfoNCE require more negative samples, the authors should at least attempt to describe the computational scaling requirements of models using InfoNCE for comparable performance or compare the computational requirements for each method to achieve what are usually small marginal advantages over other methods, so the reader can understand whether fractions of a percent of accuracy are worth additional compute. Along those lines, the authors should at least run experiments on a larger GPU mem or mini-batch (e.g., cluster GCN) experiments that run OOM.. a failed experimental implementation should not be reported in the table. The intereptation of Observation 6 is not much of an argument at all but a speculation and should be removed to strengthen the paper. Instead, adding details on the datasets used might be helpful, and esp. to ensure that accuracy alone is fair, e.g., that there are not class imbalances.


**Clarity:**

The questions that the authors aim to address are clearly stated and the paper is well written. There are certainly details that could be slightly expanded upon but leaving the details to the appendix is appropriate, given the breadth of discrete variables per design dimension, and tastefully done. It is not clear what the data is and there should at least be a table in the main text (if possible) or a quick indication of number of edges and nodes in each dataset, to ensure that the datasets chosen are represent many scales.

The experiment adding stochastic and deterministic augmentations is not clearly described and seems to do much worse than topology augmentations anyway so, for clarity, I would suggest Fig 4 and the corresponding section be deleted. The results just confuse and indicate something weird may be happening.


**Correctness:**

Some of the answers to the posed questions, e.g., "What is the most contributory component in an effective GCL algorithm" may differ for each architecture and dataset. The authors should justify why their empirical results should generalize to such a broad claim re: contrastive learning on graphs.

The problem formulation only considers graphs with node features and unweighted adjacency matrices. The authors should discuss how to incorporate and generalize their empirical study to graphs with edge features. Alternatively, they should clarify why they don't consider link prediction tasks.

Another equally valid speculation re:Table 2 results is that removing edges preserves true information, whereas adding edges adds noise. This highlights that many claims in the results are speculations and have scant theoretical justification. This would be fine if the entire space were explored, instead, the authors limit such experiments to one objective (InfoNCE) and contrasting modes. Would these results hold if a different objective were tried? To make the results more correct, specific details should be added, e.g., "Observation 1. With InfoNCE and local-local contrasting reps, ..."

In Table 2, only small graphs are used. It is incorrect to say that medium to large graphs are used, especially when OGB is available for millions of nodes.

From Fig 2, it seems that the accuracy doesn't depend on any dropout probability at all, in fact, while the authors claim that there are significant observations to be drawn from these minuscule changes. It would help if the authors add error bars to their figure and test extreme cases (why not put dropout probability closer to 1 to see decay and 0 to recover the "None" case)?


**Documentation:**

The PyGCL GitHub repo is well documented and has simple, clean, and reproducible examples for comparator models.

**Ethics:**

No concerns.

**Relation To Prior Work:**

There is at least one prior work (Design space of GNNs by Leskovec's group) that is similar in trying to understand how message passing network designs influence performance that the authors could either mention or build upon in standardizing some of the GCL methods. The authors could use that to disentangle architectural differences vs. impacts of dimensions they identify as sole "designs". In addition, section 2.2 and some of the comparisons covered in the methods are not quite extensively discussed, even though some papers, such as You et al. and the BYOL paper. This may be fine but it would be nice to see what datasets and performance each method achieves, since it remains unclear how this work relates to those benchmarks (as the authors originally constructed) or whether there are shared evaluations across GCL papers. Furthermore, some details are not explained, which should just be fixed, e.g., SP-JSD v. JSD.


**Summary And Contributions:**

PyGCL attempts to study key design dimensions in graph contrastive learning by empirically evaluating the impact of a number of design choices on the accuracy of node and graph classification tasks for  a few small graph datasets. PyGCL is also available as a GitHub repo, which allows developers to attempt several different GCL methods and try different losses, serving as a resource for algorithm development. A number of observations are made as "contributions", though these observations do not generalize as no theoretical justification is provided and expansive experimentation is not conducted. From these results, it seems InfoNCE is a consistently useful loss, although may be impractical, given its compute requirements. Topological rather than feature augmentations are beneficial, mixing local-global augmentations is not a good idea, and negative mining is not worth the effort. However, these observations are tempered by completely different results from SSL for architectures that don't require negative samples so the work motivates future study using a clean, standardized framework like PyGCL to continue studying these design dimensions.

---

> ### Author Response · Authors · 2021-09-26
> **Authors' response (1/5)**
>
> Thanks for your detailed and constructive comments! Below, we would like to make the following clarifications regarding your main concerns. Due to the length constraint on the comment box, we split our response into several parts.
>
> **Q1. Observations do not generalize as no theoretical justification is provided and expansive experimentation is not conducted; the design space is too large for the approach to study and no argument to justify that claims are general**
>
> A: Thanks for your kind suggestions. For the presented empirical studies, we believe our observations and insights are generic for graph contrastive learning for the following reasons: (1) a variety of datasets from different domains; (2) wide coverage of representative works; and (3) comprehensive experiments. Specifically, our considered datasets, spanning from social networks to chemical molecular graphs, contain with up to 35K nodes and 40K graphs for node classification, graph classification, and graph regression tasks. Then, we consider four components of graph contrastive learning. Note that these four design dimensions, if not general, are at least representative in a wide range of existing graph contrastive learning algorithms. Last, we perform extensive experiments regarding four common components and suggest a set of best practices for an effective algorithm.
>
> To ensure convincing experiments and observations, we first perform an exhaustive search over the entire design space. Then, we select and report representative results to reveal common, useful practices. We will add detailed description accordingly to strengthen the argument of our discoveries.
>
> In addition, we acknowledge that our empirical studies throw up many questions in need for further theoretical justification. However, since there has been little theoretical analysis of graph contrastive learning algorithms, we shall leave it as a future direction.
>
> **Q2. The studied graphs have fewer than 500 nodes; it is incorrect to say that medium to large graphs are used, especially when OGB is available for millions of nodes; there are many publicly available medium to large scale graphs available as benchmarks so the data examined should at least be representative of the current state of research.**
>
> A: Thanks for mentioning this problem. There might be some misunderstandings. Our work conducted experiments on both node and graph levels. Although the number of nodes per graph is less than 500 nodes, the graph-level tasks include datasets with up to 40K graphs. It should be noted that, for graph property prediction tasks, almost all [TUDataset](https://chrsmrrs.github.io/datasets/docs/datasets/) as well as [OGB datasets](https://ogb.stanford.edu/docs/graphprop/) all have less than 500 nodes per graph. In our experiments, specifically, we perform graph classification and graph regression on two OGB datasets (Appendix C.1).  Moreover, the examined datasets are widely used in recent representative models, e.g., GraphCL, GCA, InfoGraph, BGRL, GBT, GIC, etc.
>
> **Q3. There are some interesting results, e.g., that augmentations greatly improve node but not graph classification, that the authors don't comment on and in so omitting such results, fail to point out interesting trends in the field, e.g., that the design space also depends on the task.**
>
> A: Thanks for pointing out the problem. From Table 2, we observe that in general both node and graph classification benefit from data augmentation. For example, on NCI1, topology augmentation achieves up to 15% absolute improvements. We also note the degrees of improvements vary among different datasets, possibly due to the domain and the size of datasets.
>
> Regarding the design space, we agree with you that our considered design space depends on the evaluation tasks. For instance, for graph-level tasks, we have three possible contrastive modes available: global-global, global-local, and local-local, compared to node-level tasks where only two modes are involved. Thanks again for mentioning this problem; we will make this point clearer in the text.
>
> **Q4. Only graphs with node features and unweighted adjacency matrices are considered; not clear how to incorporate and generalize their empirical study to graphs with edge features.**
>
> A: Thanks for your kind suggestions. We have already included experiments on graphs that have edge features. In this case, we examine two extra augmentation functions: Edge Attribute Masking (EAD) and Edge Feature Dropout (EAD). Please refer to Appendix C.1 for experimental results and detailed discussions.

---

> ### Author Response · Authors · 2021-09-26
> **Authors' response (2/5)**
>
> **Q5. No experiments for link prediction tasks are considered.**
>
> A: Thanks for your kind advice. It should be noted that most existing graph contrastive learning work considers only node- and graph-level tasks. Due to space limitation, in our work, we consider node classification, graph classification, and graph regression tasks for evaluation. In the future, we plan to conduct additional experiments that consider more diverse downstream tasks.
>
> **Q6. Many claims in the results are speculations and have scant theoretical justification. This would be fine if the entire space were explored, instead, the authors limit such experiments to one objective (InfoNCE) and contrasting modes. Would these results hold if a different objective were tried? To make the results more correct, specific details should be added, e.g., "Observation 1. With InfoNCE and local-local contrasting reps, ..."**
>
> A: Thanks for your constructive suggestions. We actually did an exhaustive search over the design space at first. Then, we present representative results to reveal useful practices. For example, to study the impact of augmentation functions, we also experimented with the JSD objective and the global-local contrasting mode. Our results are as follows and our observations remain the same. We will add detailed description to the main text regarding this problem to make our observation more convincing.
>
>
> | Topo. aug. | Wiki       | CS         | Physics    | Computer   | NCI1       | PROTEINS   | IMDB-M     | COLLAB     |
> | ---------- | ---------- | ---------- | ---------- | ---------- | ---------- | ---------- | ---------- | ---------- |
> | None       | 67.01±0.21 | 90.35±0.21 | 92.32±0.66 | 69.32±0.53 | 58.32±2.39 | 71.01±0.98 | 45.54±1.87 | 65.88±1.23 |
> | EA         | 72.37±0.22 | 92.45±0.22 | 93.45±0.03 | 72.40±0.56 | 70.34±0.76 | 71.23±0.55 | 44.76±1.22 | 67.92±0.67 |
> | ER         | 76.02±0.31 | 91.90±0.25 | 94.21±0.09 | 76.78±0.54 | 72.88±0.54 | 72.45±0.21 | 44.89±1.32 | 67.56±0.45 |
> | EF         | 73.88±0.45 | 91.89±0.22 | 92.98±0.23 | 75.99±0.45 | 73.86±0.32 | 70.03±1.56 | 43.88±1.02 | 67.45±0.23 |
> | ND         | 77.03±0.27 | 92.00±0.45 | 93.78±0.34 | 78.03±0.43 | 72.32±1.10 | 72.32±0.65 | 47.54±0.98 | 70.32±0.45 |
> | PPR        | 69.02±0.09 | 91.65±0.12 | OOM        | 77.39±0.99 | 57.90±0.43 | 71.88±1.43 | 45.03±0.87 | 68.03±0.88 |
> | MKD        | 69.68±0.23 | 91.89±0.23 | OOM        | 70.93±0.39 | 57.83±0.65 | 70.88±0.32 | 44.99±0.90 | 67.59±0.45 |
> | RWS        | 76.34±0.29 | 92.48±0.12 | 93.88±0.15 | 74.32±0.45 | 74.88±0.99 | 71.43±0.52 | 45.03±1.23 | 70.23±0.78 |
>
>
> | Feat. aug. | Wiki       | CS         | Physics    | Computer   | NCI1       | PROTEINS   | IMDB-M     | COLLAB     |
> | ---------- | ---------- | ---------- | ---------- | ---------- | ---------- | ---------- | ---------- | ---------- |
> | FM         | 76.56±0.23 | 90.74±0.43 | 92.89±0.31 | 73.32±0.45 | 64.84±0.21 | 71.22±0.54 | 45.23±1.43 | 70.03±0.21 |
> | FD         | 76.43±0.34 | 90.66±0.28 | 92.99±0.22 | 72.98±0.32 | 63.43±0.65 | 71.77±0.87 | 45.98±0.89 | 70.12±0.89 |
>
>
> **Q7. From Fig 2, it seems that the accuracy doesn't depend on any dropout probability at all, in fact, while the authors claim that there are significant observations to be drawn from these minuscule changes. It would help if the authors add error bars to their figure and test extreme cases (why not put dropout probability closer to 1 to see decay and 0 to recover the "None" case)?**
>
> A: Thanks for your suggestions. From Figure 2, although the model performance is not sensitive to the dropout probability overall, we observe that the more sparse graph views usually lead to better performance, as long as the removal probability is not overly high. In fact, we have already added error bars in the figure but with an 70% opacity; we promise to make them clearer in the revision.
>
> As suggested, we also conduct experiments regarding two extreme cases of edge removal, as presented below. The results echo our observation that sparse graph views are preferable but the removal probability should not be overly high (N.B. a randomly initialized GCN is able to achieve ~90% classification accuracy on the CS dataset). This observation also accords with one very recent work [1, Theorem 1] which theoretically proves that a reasonable augmentation scheme should not perform aggressively such that a certain amount of information related to the downstream tasks will be kept.
>
> | Edge removal probability | Accuracy   |
> | ------------------------ | ---------- |
> | None                     | 90.76±0.05 |
> | 0.00001                  | 90.77±0.08 |
> | 0.0001                   | 90.84±0.05 |
> | 0.001                    | 91.37±0.06 |
> | 0.99                     | 91.91±0.13 |
> | 0.999                    | 91.92±0.15 |
> | 0.9999                   | 91.91±0.09 |

---

> ### Author Response · Authors · 2021-09-26
> **Authors' response (3/5)**
>
> **Q8. It is not clear what the data is and there should at least be a table in the main text (if possible) or a quick indication of number of edges and nodes in each dataset, to ensure that the datasets chosen are represent many scales.**
>
> A: Thank you for your helpful comments. We will add brief introduction of datasets along with their statistics in the main text for better readability.
>
> **Q9. The experiment adding stochastic and deterministic augmentations is not clearly described and seems to do much worse than topology augmentations anyway so, for clarity, I would suggest Fig 4 and the corresponding section be deleted.**
>
> A: Thanks for noticing this issue. Existing work employs two kinds of structural augmentations: (1) stochastic augmentations like node dropping (ND) and (2) deterministic augmentations such as Personalized PageRank (PPR) and Markov Diffusion Kernel (MDK). We also find that prior studies usually leverage a stochastic augmentation scheme after deterministic augmentations. Thus, we would like to know their **independent** impact to the performance. From the results in Table 2, we observe that using deterministic schemes solely leads to the unsatisfactory performance, barely outperforming the baseline, which further motivates us to explore the combination of stochastic and deterministic augmentation functions, as presented in Figure 4. From Figure 4, we observe that the joint scheme does improve their independent counterparts by large margins, sometimes even achieving the best performance compared to other stochastic augmentations (cf. Table 2). We will add discussions to make the motivation easier to understand.
>
> **Q10. There is at least one prior work (Design space of GNNs by Leskovec's group) that is similar in trying to understand how message passing network designs influence performance that the authors could either mention or build upon in standardizing some of the GCL methods.**
>
> A: Thanks for bringing this work to our attention. We do agree that our work shares similar motivation with Jure's work — to compare existing models at component level in a standardized setting. We will cite their work in the introduction part to make our motivation clearer.
>
> **Q11. Section 2.2 and some of the comparisons covered in the methods are not quite extensively discussed, even though some papers, such as You et al. and the BYOL paper. This may be fine but it would be nice to see what datasets and performance each method achieves, since it remains unclear how this work relates to those benchmarks (as the authors originally constructed) or whether there are shared evaluations across GCL papers.**
>
> A: Thanks for your suggestions. In fact, the implementation of prior works examined in Section 2.2 differs a lot from GNN architectures, augmentations, optimizers, tricks, etc. To see this, you may refer to [this repository](https://github.com/GraphCL/PyGCL/tree/main/examples) for several models that we reproduce using PyGCL. It is therefore difficult to directly compare our reported performance with theirs. We also note that there is one survey paper [2, Table 4] that summarizes experimental configurations and datasets of representative models. We will mention [2] in our work for readers of interest to facilitate more reproducible research.
>
> **Q12. Some details are not explained, which should just be fixed, e.g., SP-JSD v. JSD.**
>
> A: Thanks for your careful review. We empirically find that SP-JSD performs similarly to JSD. Therefore, we stick to SP-JSD in all experiments. We will make clarification in the text to avoid possible confusion.

---

> ### Author Response · Authors · 2021-09-26
> **Authors' response (4/5)**
>
> **Q13. Given the authors say the InfoNCE require more negative samples, the authors should at least attempt to describe the computational scaling requirements of models using InfoNCE for comparable performance or compare the computational requirements for each method to achieve what are usually small marginal advantages over other methods, so the reader can understand whether fractions of a percent of accuracy are worth additional compute.**
>
> A: Thank you for mentioning this problem. We have compared the memory usage of negative-sample-free and negative-sample-based approaches, as shown in Table 15. It is seen that negative-sample-free models greatly reduce the computational burden by 60%. Given that we are allowed to use one more page during revision, we will move to this experiment to the main text for better understanding the needed computational resource.
>
> Moreover, for node classification, since negative-sample-based objectives are memory-demanding, we conducted an additional experiment for negative-sample-free objectives as follows. We also cite results from recent work BGRL that leverages subsampling techniques for the InfoNCE objective to show the numbers of negative samples $k$ needed for achieving comparable performance of negative-sample-free objectives.
>
> | Objectives         | Accuracy   |
> | ------------------ | ---------- |
> | BL                 | 71.64±0.12 |
> | BT                 | 70.07±0.16 |
> | VICReg             | 71.16±0.14 |
> | InfoNCE ($k=2$)    | 60.24±4.06 |
> | InfoNCE ($k=8$)    | 70.33±0.18 |
> | InfoNCE ($k=32$)   | 71.18±0.16 |
> | InfoNCE ($k=2048$) | 71.51±0.11 |
>
> **Q14. The authors should at least run experiments on a larger GPU mem or mini-batch (e.g., cluster GCN) experiments that run OOM.. a failed experimental implementation should not be reported in the table.**
>
> A: Thanks for your feedback. On the one hand, we implemented a sparse version for both MDK and PPR, following the conventional diffusion operations, that reduces memory requirements by design. Then, we have also tried running experiments on larger V100S GPUs with 32G memory, but we still encounter the OOM error. Therefore, we choose to report OOM error which states the limitation of certain experimental configurations, so that readers could make a choice suitable for their computing resource. On the other hand, we agree with you that inductive samplers, e.g., GraphSAINT, ClusterGCN, are helpful for relieving such a problem. However, in our case, we believe these subgraph-based samplers may not be suitable for diffusion-based augmentations, considering that MDK and PPR work by enriching the graph structure with more global connectivities.

---

> ### Author Response · Authors · 2021-09-26
> **Authors' response (5/5)**
>
> **Q15. The interpretation of Observation 6 is not much of an argument at all but a speculation and should be removed to strengthen the paper. Instead, adding details on the datasets used might be helpful, and esp. to ensure that accuracy alone is fair, e.g., that there are not class imbalances.**
>
> A: The studied schemes, originally designed for grid data, measures the relative hardness of negative pairs using dot-product of embeddings. Our observation for Section 3.3 is that adopting these hard negative mining schemes naïvely for graph-structured data may end up selecting hard but false negative samples, due to the smoothing nature of GNNs. To see this clearly, we present a histogram of negatives and their semantic similarity scores with a randomly selected anchor node from the Wiki dataset. Please refer to this [link](https://www.dropbox.com/s/ulmys4e6tubcu2y/similarity.pdf) for the figure. It is evident from the figure that with the similarity increasing, there are more positive samples (i.e. false negatives) as shown in blue, possibly leading to wrong selection of hard negatives. Furthermore, at the beginning of training, node embeddings suffer from poor quality, which may be another obstacle of selecting true hard negative samples. Thanks for bringing this problem to our attention. We will add more evidence to support our observation and interpretation.
>
> Regarding evaluation metrics, for multiclass classification tasks, we measure effectiveness in terms of Micro-F1 and Macro-F1, commonly adopted in existing literature, and observe similar performance trends. Due to limited space, we only report results in terms of Micro-F1 (which is equivalent to accuracy in our case). For binary graph classification on ogbg-molhiv, we measure the effectiveness in ROC-AUC. For graph regression, we report performance in terms of MAE. As with our response to Q8, we will add brief introduction of datasets along with their statistics in the main text for better readability, given one more page during revision.
>
> Once again, we sincerely thank you for your valuable comments. We hope the above response clarifies your concerns and we are also happy to elaborate further, should you have any additional questions.
>
> References:
>
> - [1] S. Suresh, P. Li, C. Hao, and J. Neville, Adversarial Graph Augmentation to Improve Graph Contrastive Learning, 2021.
> - [2] L. Wu, H. Lin, Z. Gao, C. Tan, and S. Z. Li, Self-supervised on Graphs: Contrastive, Generative, or Predictive, 2021.

---

> ### Author Response · Authors · 2021-09-29
> **Looking forward to reply**
>
> Dear reviewer,
>
> Thanks a lot for your effort in reviewing our submission. We have tried our best to address the concerns you raised and revise our paper accordingly. It's noted that our discussion phase will end soon. It is highly appreciated if you could read our response and let us know your further questions. We are happy to clarify further.

---

> ### Author Response · Authors · 2021-09-30
> **Any comments?**
>
> Dear reviewer,
>
> Thanks a lot for your effort in reviewing our submission. We have submitted our response to your questions. Since the deadline for discussions is approaching, we are eager to know whether we have cleared your concerns.
>
> Thanks again for your consideration!

---

### Official Review · Reviewer_BSxu · 2021-09-20
**Sound paper which will help the community**

**Rating:** 7
**Confidence:** 5
**Correctness:** Good
**Clarity:** Well-written and easy to follow

**Strengths:**

The paper systematically investigate different aspects of graph contrastive learning and studies such aspects on a good number of benchmarks. The released library to investigate such models fusther is also a nice tool that can help the community. All in all, I think this paper can be a good reference point for the community.

**Weaknesses:**

The only suggestion that I have is that I wish the authors had moved beyond the TUdataset and similar ones, and had run their evaluations on the open graph benchmark (OGB). I am aware that graph CL works haven't reported contrastive results on those datasets but evaluating them on those could have been a very good baseline for future directions.

**Additional Feedback:**

NA

**Documentation:**

Well documented.

**Relation To Prior Work:**

The paper addresses the previous work.

**Summary And Contributions:**

The paper systematically studies the graph contrastive learning approaches based on Data augmentations, Contrasting modes and objectives, and Negative mining strategies. The paper does not produce any surprising results but gives a very good view over different verticals of designing GraphCL methods. They also provide an easy-to-use library featuring modularized CL components, standardized evaluation, and experiment management. This can serve the community as a good tool to further investigate the GraphCL.

---

### Author Response · Authors · 2021-09-29
**Manuscript revised: summary of changes and thanks**

We would like to thank all reviewers very much for the extensive review and constructive comments. To summarize, we are encouraged that all reviewers find our empirical study makes valuable contribution to graph contrastive learning community, is well-written and easy-to-follow, and our PyGCL library will benefit future research.

In accordance with the reviewers' comments, we have also attempted the necessary rectifications to the manuscript using the additional 10th page. In the following, we outline our changes to the paper and we highlight the changes in blue in the revised version.

- We have added details regarding the datasets, varied factors for each experiment, and implementation details.
- We present our experimental configurations more clearly and highlight that all experiments are conducted with as many hyperparameters fixed as possible.
- We elaborate on observations and analysis regarding confusing points of experiments and model formulations.

Again, we appreciate all reviewers for their valuable suggestions which enables us to further strengthen our work. We expect that our new version resolves the reviewers' main concerns. As the deadline for our interactive discussion phase is approaching, we sincerely hope reviewers would reply for further concerns and revisit the rating in light of our revision.

---

### Decision · Program_Chairs · 2021-10-09

**Decision:**

Accept

**Comment:**

The paper studies methods for contrastive learning on graphs, categorizing and performing extensive benchmarks on existing methods. They also provide an open-source package to facilitate further study of graph contrastive methods. Reviewers felt that this was an important topic to study, and that the detailed investigations and open-source library will be of use to the community. Reviewers wnP1 and rwpd initially had some concerns about details related to experimental setup and comparisons, but these were largely addressed by the comprehensive responses provided by the authors. Congratulations on having your paper accepted to the NeurIPS 2021 Datasets & Benchmarks Track! When preparing the final version of the paper, authors are strongly encouraged to take into account the issues raised by Reviewers wnP1 and rwpd.